# Exploring red cell distribution width as a potential risk factor in emergency bowel surgery—A retrospective cohort study

Michael Berry[1], Jennifer Louise Gosling[2], Rachel Elizabeth Bartlett[3], Stephen James Brett[4,5]*

1 King's Critical Care, King's College Hospital NHS Foundation Trust, London, United Kingdom, 2 Imperial School of Anaesthesia, London, United Kingdom, 3 St. Mary's Hospital Department of Anaesthesia, Imperial College Healthcare NHS Trust, London, United Kingdom, 4 Hammersmith Hospital Department of Intensive Care, Imperial College Healthcare NHS Trust, London, United Kingdom, 5 Department of Surgery and Cancer, Imperial College London, London, United Kingdom

* Stephen.brett@imperial.ac.uk

**Data Availability Statement:** All data is accessible for external researchers via the National Emergency Laparotomy Audit (https://www.nela. org.uk/NELA_Research). NELA actively encourages

## Abstract

Increased preoperative red cell distribution width (RDW) is associated with higher mortality following non-cardiac surgery in patients older than 65 years. Little is known if this association holds for all adult emergency laparotomy patients and whether it affects 30-day or long-term mortality. Thus, we examined the relationship between increased RDW and postoperative mortality. Furthermore, we investigated the prognostic worth of anisocytosis and explored a possible association between increased RDW and frailty in this cohort. We conducted a retrospective, single centre National Emergency Laparotomy Audit (NELA) database study at St Mary's Hospital Imperial NHS Trust between January 2014 and April 2018. A total of 356 patients were included. Survival models were developed using Cox regression analysis, whereas RDW and frailty were analysed using multivariable logistic regression. Underlying model assumptions were checked, including discrimination and calibration. We internally validated our models using bootstrap resampling. There were 33 (9.3%) deaths within 30-days and 72 (20.2%) overall. Median RDW values for 30-day mortality were 13.8% (IQR 13.1%-15%) in survivors and 14.9% (IQR 13.7%-16.1%) in non-survivors, p = 0.007. Similarly, median RDW values were lower in overall survivors (13.7% (IQR 13%-14.7%) versus 14.9% (IQR 13.9%-15.9%) (p<0.001)). Mortality increased across quartiles of RDW, as did the proportion of frail patients. Anisocytosis was not associated with 30-day mortality but demonstrated a link with overall death rates. Increasing RDW was associated with a higher probability of frailty for 30-day (Odds ratio (OR) 4.3, 95% CI 1.22–14.43, (p = 0.01)) and overall mortality (OR 4.9, 95% CI 1.68–14.09, (p = 0.001)). We were able to show that preoperative anisocytosis is associated with greater long-term mortality after emergency laparotomy. Increasing RDW demonstrates a relationship with frailty. Given that RDW is readily available at no additional cost, future studies should prospectively validate the role of RDW in the NELA cohort nationally.

secondary research and provides both aggregate and patient level data. Data for this research is available fully anonymised and de-identified.

**Funding:** This research received no specific grant from any funding agency in the public, commercial or not-for-profit sectors. Infrastructure support for this research was provided by the NIHR Imperial Biomedical Research Centre (BRC). The views are those of the author(s) and not necessarily those of the NIHR or the Department of Health and Social Care.

**Competing interests:** MB was a Health Service Research Centre fellow with the National Emergency Laparotomy Audit from August 2018 to August 2019. All other authors (JLG, REB, SJB) declare that they have no conflict of interest. We received no financial support for the research, authorship or publication of this article other than Imperial Biomedical Research Centre infrastructure support for Professor Brett. This does not alter our adherence to PLOS ONE policies on sharing data and materials.

## Introduction

Every year, approximately 24,000 emergency laparotomies are performed across England and Wales. Postoperative mortality remains high, especially in older patients with comorbidities [1]. Determining surgical risk accurately for individual patients is essential and increasingly emphasised yet remains challenging.

Numerous models have been developed to guide decision making and allow comparison of surgical outcomes following emergency laparotomy. In the United Kingdom, the Portsmouth Physiological and Operative Severity Score for the enUmeration of Mortality and Morbidity (P-POSSUM) model, the Surgical Outcome Risk Tool (SORT) and the National Emergency Laparotomy Audit (NELA) risk model are particularly popular [2].

Despite widespread use, risk prediction tools often have substantial limitations, including resource intensive calculations, dependence on postoperative data and validation bias [2, 3]. Consequently, an ongoing interest remains in identifying new predictors as well as developing more accurate risk prognostication models.

Recent research shows that both frailty and red cell distribution width (RDW) are significant variables in the perioperative setting [4, 5]. To date, neither have been routinely incorporated into surgical risk assessment tools.

The exact link between an elevated RDW and mortality remains unclear but is thought to denote bone marrow dysfunction, systemic inflammation and oxidative stress. Inflammatory pathways mediated by cytokines such as TNF-$\alpha$ and IL-6 inhibit erythropoietin-induced red blood cell maturation and may offer one possible explanation [6].

Importantly, emerging data suggest a strong correlation between anisocytosis, which is reported quantitatively as RDW, and mortality in the older population, perhaps reflecting the multiple physiological impairments related to ageing and frailty [7]. Therefore, RDW may serve as a marker of prior frailty and consequent mortality risk following emergency bowel surgery. Given the availability and the routine reporting of RDW as part of the full blood count, understanding its prognostic value could be both cost-effective and useful for surgical risk stratification in emergency laparotomy patients.

Using our institution's NELA dataset, we set out to answer three specific questions. First, we examined whether pre-operative RDW values on average are different between emergency laparotomy survivors and non-survivors. Second, we investigated if RDW is a useful predictor of mortality in emergency laparotomy patients and its potential additive value to the NELA model. Finally, we sought to explore whether RDW is independently associated with frailty in this population.

## Methods

### Data source, patients and outcome measures

This study was a retrospective, single-centre, clinical database analysis conducted at a tertiary London university hospital. Ethical approval for this study was agreed prospectively by the Imperial College London and Imperial College Healthcare NHS Trust Joint Research Compliance Office as well as the Health Research Authority (institutional reference number: 18SM4441/IRAS ID: 242302; HRA: 18/HRA/1860). Under prevailing United Kingdom regulations, individual patient consent was not required given the retrospective nature of the study. Data were pseudo anonymised using the unique NELA identifier. Handling of online NELA data entered by individual NHS trusts adheres to strict information governance standards, which are laid out on the NELA website [8]. All additional administrative or clinical data required were collected as part of routine clinical care and analysed in accordance with the General Data Protection Regulation.

We reviewed the St Mary's Imperial College Healthcare NHS Trust online NELA database for patients aged eighteen or older who underwent emergency laparotomy between 1st January 2014 and 31st January 2018. The follow-up period ended three months after the data accrual period on 30th April 2018. Our inclusion criteria mirrored those published by NELA [9]. Only the outcome of the index surgery was evaluated if a patient underwent multiple emergency laparotomies during their admission. Patients with no documented operative indication, date of procedure or full blood count were excluded.

Outcomes were 30-day mortality, overall mortality during the follow-up period and frailty. We defined thirty-day mortality as death occurring within 30 days of the index operation. Overall mortality was taken to mean mortality status on 30th of April 2018.

Pre-operative frailty was pragmatically evaluated. We examined the admission clerking of all patients for a documented assessment of frailty using any validated frailty measurement tool. In the absence of such an assessment, the history recorded in the admission clerking was reviewed and compared against the Clinical Frailty Scale (CFS) [10]. Scores greater than or equal to five were taken as frail, which has been shown in the literature to be associated with increased complications as well as mortality [11].

## Patient and public involvement

Patients and the public were not involved in the study.

## Data collection, missing values and predictor selection

Clinical measurements, comorbidities and expected operative findings were recorded pre-operatively. ASA grade (American Society of Anaesthesiologist physical status classification system) and operative urgency according to the National Confidential Enquiry into Patient Outcome and Death were also included. We classified operative severity according to NELA as major or major+, reflecting surgical immediacy, post-operative length of stay or associated mortality [1, 2].

Blood tests were carried out pre-operatively in our institution's laboratory and comprised haemoglobin, RDW, white blood cell count, creatinine, urea, sodium and potassium. Full blood counts were measured using the Abbot Alinity-HQ (Abbott, IL, USA) analyser. Creatinine, urea and electrolytes were determined using the Abbott Architect c8000 system (Abbott, IL, USA).

To avoid confounding interventions such as blood transfusions, which could alter the RDW, we understood pre-operative to mean the first set of blood results on admission and not immediately pre-surgery as recorded by NELA. Rarely did in-patients admitted for non-general surgical reasons need an emergency laparotomy. For this small cohort, laboratory values twenty-four hours before surgery were used. Missing NELA database values were cross-referenced with the institution's clinical information system generating a complete pre-operative dataset.

Candidate risk factors for our mortality analyses were selected *a priori* based on availability, previous reviews of existing prediction models, national guidelines and research team consensus [2, 12–15]. Thus, the following variables were included: RDW, NELA risk prediction score, haemoglobin, creatinine, and indication for surgery. The NELA risk score incorporates routinely collected predictors (patient demographics, physiological data, laboratory values, and operative details) and has been published elsewhere [2]. A full overview of the included variables can be found in the S1 Annex. Our frailty model contained the covariates sex, age, RDW and haemoglobin.

## Statistical methods and model development

We examined baseline patient characteristics across RDW quartiles and checked continuous variables for normality by plotting the data as well as using the Shapiro-Wilk test. Analysis of continuous, non-parametric data was performed using the Wilcoxon-Mann-Whitney test or the Kruskal-Wallis test as appropriate. For categorical variables, the $\chi^2$ test or Fisher's exact test were used to compare frequencies. Associations with $P$ values <0.05 were considered statistically significant.

Using RDW as a continuous variable, we went on to evaluate the prognostic value of RDW at predicting mortality outcomes. Thus, we built two separate nested multivariable Cox regression models (30-day mortality and overall mortality) using the established predictors. Comparing the reduced model (without RDW) with the full model (with RDW) using the likelihood ratio $\chi^2$ test allowed us to determine the added predictive value of RDW. Furthermore, the relative importance of RDW in the models was established using an analysis of the variance, allowing for interactions and non-linear effects.

In developing our survival models, it was necessary to combine some of the operative categories with too few patients. We regrouped the variables 'Colitis' and 'Ischaemia' with the variable 'Other'. All continuous risk factors had outliers at one end of their distribution. Therefore, the distributions were winsorised at the $5^{th}$ or $95^{th}$ percentile as required (see Table 1 of the S1 Annex). Continuous variables were assessed for non-linearity and transformed accordingly. Moreover, several clinically plausible interactions were considered and included if found to be statistically significant. We also checked both models for the proportional hazards assumption and examined for multicollinearity as well as influential observations.

Internal validation of the models was performed using bootstrap resampling, allowing us to estimate the amount of overfitting. The least absolute shrinkage and selection operator (LASSO) method was then employed to shrink regression coefficients. The updated LASSO models enabled us to draw hazard ratio charts presenting point and interval estimates of predictor effects as well as nomograms.

Finally, to investigate the association between RDW as a continuous variable and frailty, we developed a binary logistic regression model. Here we considered RDW as if it were a new diagnostic marker, aiming to characterise its relationship with frailty. We defined frailty to be dichotomous (frail or not frail) and adjusted our model for sex, age as well as haemoglobin. Using approaches similar to the ones outlined above, we checked the underlying model assumptions and penalised our regression analysis for overfitting. Missing data were examined for patterns of missing values and a complete case analysis was carried out.

Publications by Harrell, Spiegelhalter, Pavlou and Torisson informed all modelling algorithms. In designing our models, we adhered to the TRIPOD reporting guidelines [16–20]. A detailed account of their development can be found in the S1 Annex. All statistical analysis was carried out using R v3.5.2 (R Foundation for Statistical Computing, Vienna, Austria, https://www.R-project.org/) and the full code is published on GitHub(https://www.github.com/U601648/RDW_mortality_project).

## Results

Overall, 372 emergency laparotomies were recorded during the study period. Sixteen operations were excluded from the final analysis (Fig 1). Baseline participant characteristics are shown by quartiles of RDW in Table 1. For most patients, the laboratory values on admission were used, therefore minimising iatrogenic confounding. However, for fourteen in-patients

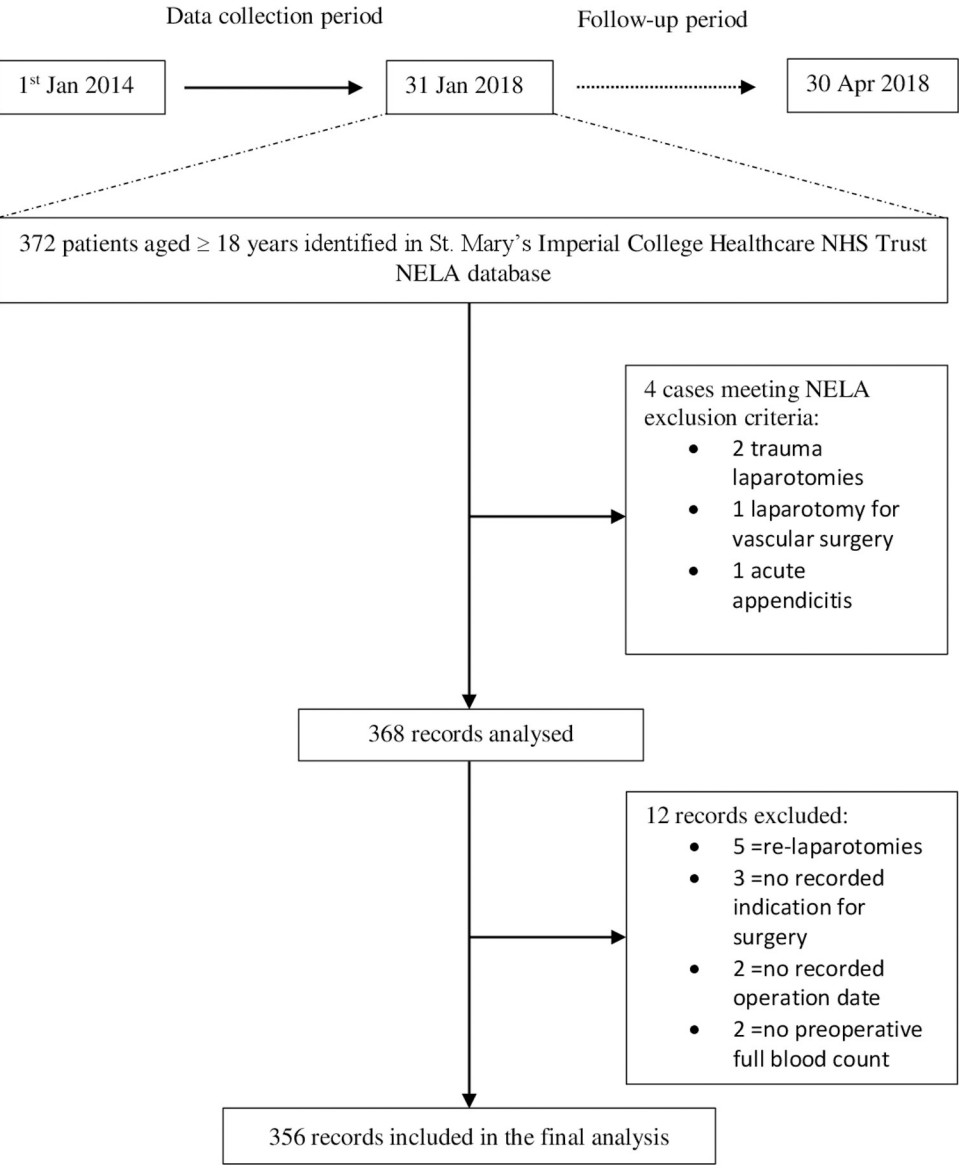

**Fig 1. CONSORT diagram of patient enrolment.**

(3.9%) requiring a laparotomy unrelated to their initial admission, blood tests twenty-four hours before surgery were utilised.

All-cause 30-day mortality was 9.3% (n = 33), while overall mortality rose to 20.2% (n = 72) after emergency bowel surgery for the total follow-up period. In those patients who died at 30-days compared to those who survived median RDW levels were consistently higher, 14.9% (IQR 13.7%-16.1%) and 13.8% (IQR 13.1%-15%) respectively, $P = 0.007$. Median RDW for overall mortality was 13.7% (IQR 13%-14.7%) in survivors versus 14.9% (IQR 13.9%-15.9%) in non-survivors, $P<0.001$. The cumulative mortality rate increased across RDW quartiles for both follow-up periods and is displayed in Fig 2.

At 30-days RDW was the least significant predictor. The relative importance of RDW improved considerably for the longer-term mortality model (Table 3 of the S1 Annex). RDW added prognostic value only for the total follow-up period with a calculated percentage of new

**Table 1. Baseline characteristics of patients undergoing emergency laparotomy across red cell distribution width (RDW) quartiles.**

| Variable | RDW quartiles | | | | |
| --- | --- | --- | --- | --- | --- |
| | 1st quartile | 2nd quartile | 3rd quartile | 4th quartile | |
| | RDW1 ≥11.7% & <13.1% (n = 93) | RDW2 ≥13.1% & <13.9% (n = 90) | RDW ≥13.9% & <15.1% (n = 86) | RDW≥ 15.1% & ≤27.3% (n = 87) | *P* value |
| **Demographic** | | | | | |
| Age | 50.0 (37–66) | 59.5 (44–76.5) | 66.0 (51–77) | 64.0 (47–74.5) | <0.001 |
| Female sex (%) | 40 (43.0) | 48 (53.3) | 44 (51.2) | 52 (59.8) | 0.159 |
| Surgical | | | | | <0.001 |
| Obstruction (%) | 45 (48.4) | 43 (47.7) | 43 (50.0) | 43 (49.4) | |
| Sepsis (%) | 35 (37.6) | 36 (40.0) | 32 (37.2) | 35 (40.2) | |
| Ischaemia (%) | 7 (7.8) | 5 (5.5) | 3 (3.5) | 5 (5.7) | |
| Haemorrhage (%) | 2 (2.2) | 2 (2.2) | 3 (3.5) | 4 (4.6) | |
| Colitis (%) | 3 (3.2) | - | 2 (2.3) | - | |
| Other (%) | 1 (1.1) | 4 (4.4) | 3 (3.5) | - | |
| **Pre-operative** | | | | | |
| Median NELA 30-day predicted mortality % | 1.2 (0.5–4.9) | 2 (0.5–9.3) | 4.7 (1.1–11.9) | 4.1 (1.75–14.0) | <0.001 |
| ASA score | | | | | 0.002 |
| ASA 1 (%) | 17 (18.3) | 17 (18.9) | 8 (9.3) | 10 (11.5) | |
| ASA 2 (%) | 45 (48.4) | 34 (37.8) | 25 (29.1) | 19 (21.8) | |
| ASA 3 (%) | 22 (23.7) | 23 (25.6) | 31 (36.0) | 32 (36.8) | |
| ASA 4 (%) | 7 (7.5) | 14 (15.6) | 21 (24.4) | 24 (27.6) | |
| ASA 5 (%) | 2 (2.2) | 2 (2.2) | 1 (1.2) | 2 (2.3) | |
| Urgency of surgery | | | | | 0.123 |
| Expedited >18 hours (%) | 12 (12.9) | 12 (13.3) | 23 (26.7) | 19 (21.8) | |
| Urgent 6–18 hours (%) | 39 (41.9) | 31 (34.4) | 31 (36.0) | 31 (35.6) | |
| Urgent 2–6 hours (%) | 36 (38.7) | 42 (46.7) | 24 (27.9) | 34 (39.1) | |
| Immediate <2 hours (%) | 6 (6.5) | 5 (5.9) | 8 (9.3) | 3 (3.4) | |
| ECG | | | | | 0.601 |
| No abnormalities (%) | 85 (91.4) | 76 (84.4) | 78 (90.7) | 78 (89.7) | |
| AF rate 60–90 min$^{-1}$ (%) | 2 (2.2) | 6 (6.7) | 4 (4.7) | 2 (2.3) | |
| AF rate >90 min$^{-1}$ or any other abnormal rhythm, ST changes (%) | 6 (6.5) | 8 (8.9) | 4 (4.7) | 7 (8.0) | |
| Cardiac signs | | | | | 0.826 |
| No failure (%) | 80 (86.0) | 71 (78.9) | 72 (83.7) | 67 (77.0) | |
| Diuretic, digoxin, antianginal or hypertensive therapy (%) | 10 (10.8) | 14 (15.6) | 11 (12.8) | 17 (19.5) | |
| Peripheral oedema, warfarin therapy (%) | 2 (2.2) | 2 (2.2) | 1 (1.2) | 2 (2.3) | |
| Raised JVP or CXR signs (%) | 1 (1.1) | 3 (3.3) | 2 (2.3) | 1 (1.1) | |
| Respiratory history | | | | | 0.611 |
| No dyspnoea (%) | 77 (82.8) | 72 (80.0) | 67 (77.9) | 70 (80.5) | |
| Dyspnoea on exertion (%) | 11 (11.8) | 9 (10.0) | 16 (18.6) | 10 (11.5) | |
| Dyspnoea limiting exertion (%) | 3 (3.2) | 6 (6.7) | 2 (2.3) | 6 (6.9) | |
| Dyspnoea at rest (%) | 2 (2.2) | 3 (3.3) | 1 (1.2) | 1 (1.1) | |
| Clinical values | | | | | |
| Haemoglobin (gl$^{-1}$) | 143 (133–151) | 139 (125–148) | 125 (113–139) | 120 (96–132) | <0.001 |
| Creatinine (μmoll$^{-1}$) | 76 (67–92) | 73(64–101.8) | 76 (65–102) | 79 (65.5–113) | 0.828 |
| Urea (mmoll$^{-1}$) | 5.5 (4.4–7.5) | 5.8 (3.6–9.0) | 6.1 (4.2–9.0) | 6 (4.1–9.35) | 0.876 |
| Sodium (mmoll$^{-1}$) | 138 (135–139) | 139 (136–141) | 138 (135–139) | 137 (135–140) | 0.059 |
| WBC (x10$^9$l$^{-1}$) | 12.2 (8.9–17.3) | 10.4 (7.3–13.4) | 9.9 (8.0–14.2) | 9.9 (6.0–13.8) | 0.051 |

*(Continued)*

**Table 1.** (Continued)

| | RDW quartiles | | | | |
|---|---|---|---|---|---|
| **Variable** | **1st quartile** | **2nd quartile** | **3rd quartile** | **4th quartile** | |
| | **RDW1 $\geq$11.7% & <13.1% ($n$ = 93)** | **RDW2 $\geq$13.1% & <13.9% ($n$ = 90)** | **RDW $\geq$13.9% & <15.1% ($n$ = 86)** | **RDW$\geq$ 15.1% & $\leq$27.3% ($n$ = 87)** | **$P$ value** |
| Systolic blood pressure (mmHg) | 129 (113–140) | 122 (109–138) | 124 (107–134) | 122 (108–134) | 0.484 |
| Pulse (beats min$^{-1}$) | 86 (75–101) | 88 (75–102) | 84 (76–95) | 88 (80–108) | 0.204 |
| **Perioperative** | | | | | |
| Operative severity | | | | | 0.922 |
| Major (%) | 60 (64.5) | 56 (62.2) | 58 (67.4) | 56 (64.4) | |
| Major+ (%) | 33 (35.5) | 34 (37.8) | 28 (32.6) | 31 (35.6) | |
| Peritoneal soiling | | | | | 0.826 |
| None (%) | 44 (47.3) | 44 (48.9) | 46 (53.5) | 33 (37.9) | |
| Serous fluid (%) | 20 (21.5) | 17 (18.9) | 24 (27.9) | 20 (23.0) | |
| Localised pus (%) | 5 (5.4) | 4 (4.4) | 4 (4.7) | 6 (6.9) | |
| Free bowel content, pus, or blood (%) | 24 (25.8) | 25 (27.8) | 12 (14.0) | 28 (32.2) | |
| Intraoperative blood loss | | | | | 0.812 |
| <100ml (%) | 32 (34.4) | 29 (32.2) | 26 (30.2) | 35 (40.2) | |
| 101-500ml (%) | 54 (58.1) | 56 (62.2) | 54 (62.8) | 47 (54.0) | |
| 501-999ml (%) | 5 (5.4) | 3 (3.3) | 5 (5.8) | 4 (4.6) | |
| >1000ml (%) | 2 (2.2) | 2 (2.2) | 1 (1.2) | 1 (1.1) | |
| Severity of malignancy | | | | | 0.826 |
| None (%) | 85 (91.4) | 77 (85.6) | 71 (82.6) | 67 (77.0) | |
| Primary only (%) | 1 (1.1) | 5 (5.6) | 10 (11.6) | 11 (12.6) | |
| Nodal metastases (%) | 1 (1.1) | 0 (0) | 4 (4.7) | 2 (2.3) | |
| Distant metastases (%) | 6 (6.5) | 8 (8.9) | 1 (1.2) | 7 (8.0) | |
| Observed 30-day mortality (%) | 4 (4.3) | 6 (6.7) | 10 (11.6) | 13 (14.9) | 0.061 |
| Observed overall mortality (%) | 8 (8.6) | 12 (13.3) | 20 (23.3) | 32 (36.8) | <0.001 |

Continuous variables are shown as median and interquartile ranges. Categorical variables are shown as a frequency (%). Non-winsorised values were used to draw up the table. P values were calculated using the Kruskal-Wallis test for continuous variables and $\chi^2$ test/Fisher's exact test was used for categorical data (testing for overall difference in RDW quartiles). Obstruction (= small & large bowel obstruction), sepsis (= peritonitis, abdominal abscess, perforation, anastomotic leak), ischaemia (= small & large bowel ischaemia), other (= abdominal compartment syndrome, swallowed foreign body, wound dehiscence, seroma). AF: atrial fibrillation, ASA: American Society of Anaesthesiologist physical status classification system, CXR: chest radiograph, ECG: electrocardiogram, JVP: jugular venous pulse, Major+: all colonic resections, gastrectomy, laparostomy, intestinal bypass, reoperations for bleeding/sepsis, Major: all other including stoma formation, small bowel resection, adhesiolysis, repair of perforated/bleeding ulcer, NELA: National Emergency Laparotomy Audit, RDW: red cell distribution width, WBC: white blood cell count.

information of 14%. The overall mortality Cox regression model was internally validated via bootstrapping (1000 resamples) to penalise for possible overfitting. The likely future predictive discrimination measured by Somers' $D_{xy}$ rank correlation is 0.46 for the base model and 0.50 for the full model. Optimism adjusted C-statistics were 0.73 and 0.75, respectively. The estimated slope shrinkage was 0.82 and 0.83, suggesting that approximately 0.18/0.17 of the model fitting is noise, especially with regard to calibration accuracy implying moderate overfitting. The calibration curve for the full model is shown in the Fig 6 of the S1 Annex.

LASSO regression was used to shrink regression coefficients and revise the full overall mortality model (mean shrinkage 1.02). To present point and interval estimates of the updated predictor effects a hazard ratio chart was plotted alongside a nomogram for predicting death in emergency laparotomy patients over the total follow-up period (Fig 3).

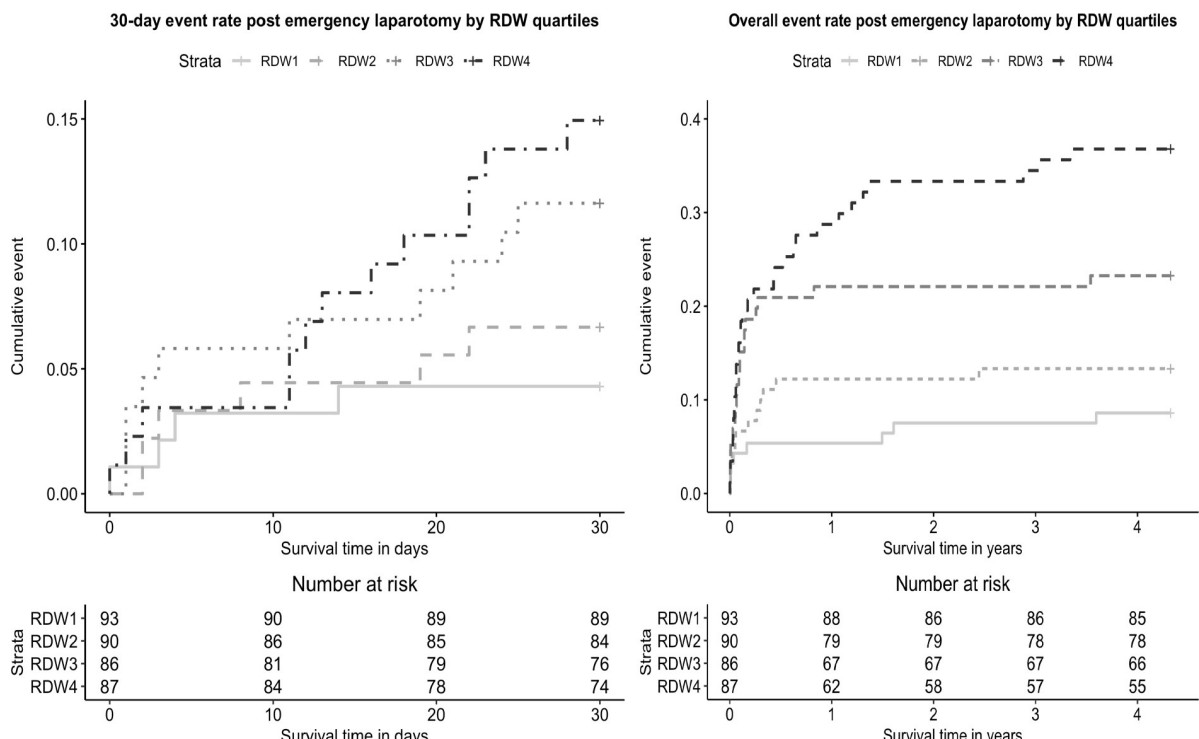

**Fig 2. Cumulative mortality rate plots for 30-day and overall mortality post emergency laparotomy by RDW quartiles.** The log-rank test was significant for the total follow-up period $\chi^2$ (log-rank) = 25.5, d.f. = 3, p<0.001 (d.f. degrees of freedom). For 30-day mortality the survival lines cross and the log-rank test is unlikely to detect a difference and should not be used for methodological reasons [21].

Assessment of frailty was often not recorded, making any judgement about frailty problematic. Hence, it was only possible to draw valid conclusions regarding frailty in 140 (39.3%) patients. Of these, 26 (18.5%) had a formal assessment of frailty documented. All other frailty data (114, 81.5%) were gathered from patient records. Baseline descriptive statistics for the cohort are presented in Table 6 of the S1 Annex.

A significantly higher proportion of patients that died at 30-days were frail (Odds ratio (OR) 4.3, 95% CI 1.22–14.53, $P$ = 0.01). Similarly, the risk of frailty was higher amongst patients that died overall (OR 4.9, 95% CI 1.68–14.09, $P$ = 0.001). Comparing the cohort across groups of RDW demonstrated a higher proportion of frail individuals in each progressive quartile (RDW1: 2 (n = 39), RDW2: 3 (n = 37), RDW3: 7 (n = 32), RDW4: 12 (n = 32)) and was statistically significant, $\chi^2$(3, N = 140) = 15.9, p = 0.001.

Based on binary logistic regression analysis, pre-operative RDW was independently associated with frailty in our emergency laparotomy cohort. Validating our model using 400 bootstrap replications the bias-corrected estimate of predictive discrimination was $D_{xy}$ = 0.462 (C-static 0.73). The corrected Brier score was 0.134, and the estimated maximum calibration error in predicting frailty was 0.06 (Table 8 of the S1 Annex). We depicted the fitted model by computing odds ratios with their respective 95% confidence intervals and graphed the association of RDW with frailty in NELA patients, estimated for a range of different ages (Fig 4).

## Discussion

To our knowledge, this study is the first to examine pre-operative RDW and mortality, its potential added predictive value, and its relationship with frailty in emergency laparotomy

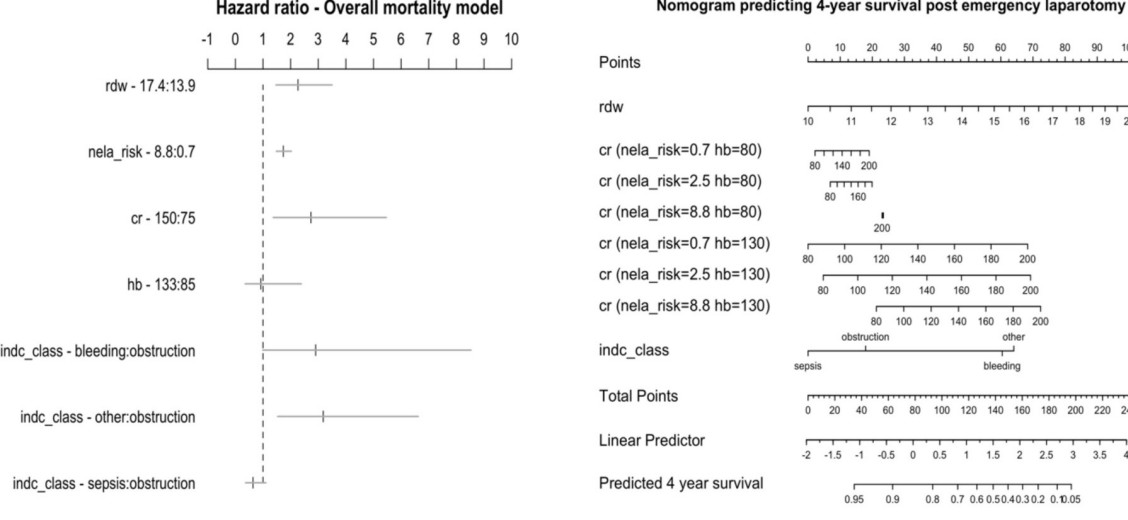

**Fig 3. Hazard ratio chart and nomogram for overall mortality post emergency laparotomy.** Top panel: Estimated hazard ratios (HR) and 95% confidence bars for the overall mortality model. For the NELA risk score interquartile range HR are used, for all other continuous predictors median values are compared to the 90th (RDW, creatinine) or 5th centile (haemoglobin). For example, when RDW changes from its median value (13.9%) to the 90th centile (17.4%), the hazard ratio more than doubles (HR 2.3, 95% CI 1.5–3.5). Standard HRs are presented for surgical indication. Here the hazard ratio is a conventional comparison of the hazard between two groups. Bottom panel: Nomogram for predicting all-cause mortality following emergency laparotomy for the total follow-up period. For each predictor, determine the points assigned on the 0–100 scale and add those points. Plot the result on the Total Points scale and then read the corresponding predictions below it. The linear predictor of a Cox model is a weighted sum of the variables in the models, where the weights are the regression coefficients. Note the effect of interactions, the risk of creatinine is influenced by haemoglobin and the NELA risk score. To illustrate this the 5th and 90th centile was chosen for haemoglobin and the interquartile range for the NELA risk score. RDW: Red cell distribution width (%), hb: haemoglobin (gl⁻¹), cr: creatinine (µmoll⁻¹), nela_risk: NELA risk score, indc_class: indication for laparotomy.

patients. We found that RDW values, on average, were higher in non-survivors. Moreover, there was a distinct gradient in overall mortality risk associated with increasing RDW. This association remained after accounting for the NELA risk score, haemoglobin, creatinine and operative indication for overall mortality but not shorter-term 30-day mortality.

In the peri-operative setting, anisocytosis has been mainly associated with long-term mortality after surgery [22]. However, more recently, Abdullah and colleagues described a link between 30-day mortality and pre-operative RDW in patients 65 years or older undergoing noncardiac surgery [4]. This differs from our findings and others, where RDW was not a convincing predictor of death at 30-days [23, 24]. In line with a recently published retrospective database study, RDW had a stronger association with overall mortality in our emergency laparotomy cohort [24]. A discrepancy, which is likely to have arisen due to differences in study population and methodology. For example, we did not dichotomise RDW using sensitivity analyses but explored RDW as a quantitative variable avoiding the categorisation of an inherently continuous marker. Furthermore, the choice of regression coefficients is likely to account for much of the observed disparity. Our findings show that the composite NELA risk score is the main predictor of all-cause mortality in both models. The NELA tool was developed to produce risk-adjusted 30-day postoperative mortality rates [2]. Thus, RDW is probably not influential enough in our model at 30-days, lacking in discriminatory power and adding little in predictive value compared with the NELA risk score. Conversely, the NELA risk score was

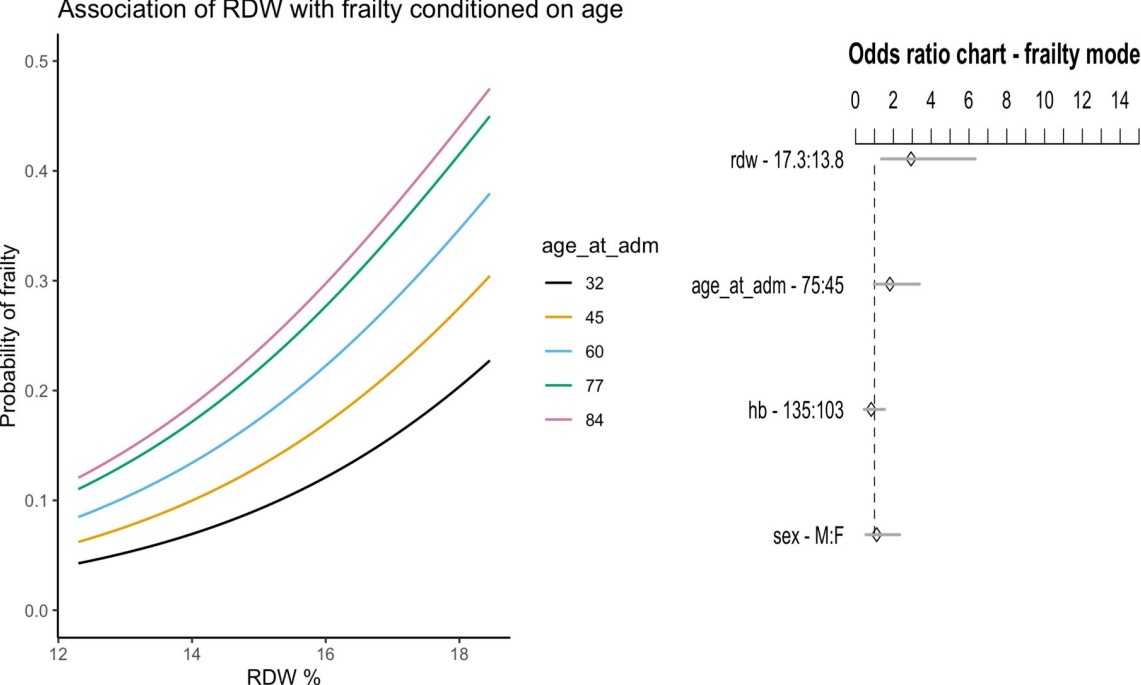

**Fig 4. Frailty logistic regression model.** The left-hand panel displays an estimated odds ratio (OR) chart and respective 95% confidence intervals. For example, when RDW changes from the 50th to the 90th percentile (13.8% to 17.3%) the odds ratio of being frail is 2.9 (95% CI 1.4–6.4). The odds for age (OR 1.8, 95% CI 1–3.4) are for the 25th and 75th percentile, while for haemoglobin (OR 0.8, 95% CI 0.4–1.5) they are based on the 10th and 50th percentile. The right-hand panel illustrates the effect of RDW on the probability of frailty for emergency laparotomy patients, estimated for different ages. The age cut-offs represent the 10th, 25th, 50th, 75th and 90th percentile (n = 140). RDW: red cell distribution width (%), age_at_adm: age at admission (years), hb: haemoglobin (gl⁻¹).

not designed with long-term mortality in mind and may explain the improved prognostic influence of RDW on our overall model.

Though we did not expressly investigate how prognostic factors impact outcome over time, it is biologically plausible that markers differ in their predictive ability in a time-dependent manner. The NELA model primarily reflects perioperative events, which may have less influence on patients who survive long-term, usually because of treatment with curative intent. Hence, the predictive worth of the NELA risk score is likely to decrease with time after laparotomy. In contrast, pre-operative anisocytosis may indicate chronically reduced physiological reserve, making it possibly a better indicator of longer-term mortality [25]. An interesting follow-up study would be to formally evaluate at what time point RDW, as a prognostic marker, has the most significant impact on mortality prediction.

Although numerous studies, including ours, have shown an association between higher RDW and decreased survival, the exact causal relationships remain elusive and are likely to be multifactorial [4, 24, 25]. Various hypotheses have been suggested, all of which involve systemic factors that alter erythrocyte physiology, such as oxidative stress, inflammation, malnutrition and telomere length [25, 26]. Nonetheless, there is an emerging consensus that anisocytosis reflects profound physiological dysregulation.

While many of the above mechanisms are likely to be similar to those implicated in the pathophysiology of anaemia, we found anisocytosis to be independent of haemoglobin concentration. This is in keeping with findings published by Patel and colleagues [26]. Equally, haemoglobin concentration was not a meaningful predictor of all-cause mortality in our study.

A similar conclusion was reached during the development of the NELA risk prediction tool, leading to its exclusion from the model [2].

RDW is also strongly associated with advancing age and a higher disease burden [27]. More recently, a connection between RDW and frailty has been suggested, an association that we were able to support in our explorative analysis [27]. Intriguingly, frailty and anisocytosis appear to share similarities in their proposed pathophysiological mechanism [25, 28]. Thus, RDW is a possible integrative biomarker reflecting the multiple biological impairments related to increasing frailty and indirectly ageing, perhaps thereby explaining its additional predictive worth.

We acknowledge several limitations, including the single-centre, retrospective observational design of the study and its relatively small sample size, restricting its overall generalisability. While national inclusion criteria mitigate selection bias, our findings ideally require prospective confirmation across the whole NELA cohort. At present, the NELA project does not routinely collect RDW. Since RDW is easily measured as part of the full blood count, including it prospectively in large nationally or internationally collected datasets may validate its effectiveness and offer valuable insights prognostically.

A further shortcoming is that we did not account for blood transfusions, which could modify RDW. We used admission blood tests to attenuate the confounding risk of perioperative blood transfusions, but a small proportion of patients underwent laparotomy as in-patients. In an attempt to adjust for additional risk factors, we applied the amalgamated NELA score. Nevertheless, we cannot exclude the possibility of residual confounding. In particular, we did not account for nutritional deficiencies (folate, cobalamin, iron) and cancer, similar to many studies on RDW. However, a large community-based study in the United States examining RDW in middle-aged and older adults found RDW to predict mortality independent of these confounding factors [7].

Moreover, we developed our frailty model, excluding a large number of patients with missing data. The distribution of variables across risk factors was similar in patients with complete and missing frailty outcomes, suggesting that the data were missing at random (Table 7 of the S1 Annex). Reassuringly, the prevalence of frailty in our cohort mirrored a national multicentre study specifically examining frailty in NELA patients [29]. In the majority of patients, frailty was established using the clinical notes. Admittedly subjective, the simplicity of the CFS score facilitates this and is thought to be appropriate in the literature [11].

Lastly, we recognise that internal validation demonstrated overfitting for both our models. This is most likely due to the high number of parameters, including screening for non-linear terms and global interaction tests. However, our models were exploratory and not meant to be new parsimonious prediction tools. Thus, we emphasised the inclusion of clinically relevant variables alongside interactions/non-linear terms in the trade-off with overfitting [19].

Our study also had various strengths, specifically minimal loss of predictor values, *a priori* choice of covariates and a robust approach to model development. We used advanced methods to address non-linearity, interactions, internal validation and presented our models graphically with these complexities in mind. Importantly, we avoided the categorisation of RDW and many of its associated problems [11]. Some of these include the heterogeneity of diagnostic and prognostic cut-offs in the literature and unmet standardisation of erythrocyte sizing [7]. Crucially, specific cut off values imply that the relationship with an outcome is flat on either side of the chosen value, which biologically is seldom plausible [30]. Indeed, we were able to demonstrate that mortality increases across what is considered the normal range of RDW, representing a continuum of risk and is depicted in our nomogram (Fig 4).

A key strength of investigating RDW is its availability at no additional cost since it is routinely performed as part of the full blood count. Similarly, use of the CFS to screen for frailty is straightforward and uses readily available clinical information. While concerns around its

applicability in patients below 65 years of age exist, it has been used successfully in adult emergency surgical admissions regardless of age [11].

Despite mounting evidence that anisocytosis is associated with increased long-term mortality following surgery, large-scale prospective studies are now needed to validate its predictive utility [4, 24]. Going forward, investigators should focus on RDW as a continuous variable to develop valid prediction models rather than classification tools based on subjective thresholds. Moreover, these studies should now assess the added predictive value of RDW to determine if pre-operative anisocytosis enhances current risk-stratification tools. In turn, superior risk prediction tools could allow more meaningful informed consent and shared decision making between patients and healthcare professionals.

At present, it remains unknown whether RDW is a *modifiable* risk factor perioperatively, including the elective setting. It would be interesting to establish if targeting factors reflected in the RDW improves surgical outcomes. Should tailored interventions such as physical rehabilitation, nutritional support or immunomodulation prove beneficial, this would further strengthen the argument to use RDW to identify individual patients at risk [24].

Conversely, the idea that frailty contributes to increased mortality following emergency surgery is not new, nor is the concept of integrating frailty into surgical risk assessment [5]. However, whether increased RDW, as a measure of biological vulnerability, offers a valid link with frailty should now be formally investigated.

Finally, pre-operative risk models for emergency laparotomies are based on retrospective database analyses of patients undergoing surgery [2]. We know little about patients who met the criteria for surgery but did not proceed due to personal choice or perceived high risk [31]. Future research must establish the predictive value of RDW for all patients with or without surgical intervention to understand its pre-operative worth fully.

## Conclusions

We established that anisocytosis as reflected in the RDW value is associated with higher rates of postoperative mortality following emergency laparotomy. Furthermore, our analysis tentatively supports the notion that increased RDW is a possible marker of physiological dysregulation relevant to frailty [24]. While further research is needed to understand these mechanisms fully, RDW seemingly provides prognostic information that could inform future risk prediction tools. Accordingly, we explored how to quantify the added prognostic value of RDW without resorting to categorisation. Although oversimplified for illustration, our models demonstrated a statistically efficient way to investigate the relative merit of RDW. However, whether adding RDW as a global marker of homeostasis to surgical prognostication tools will improve patient management and outcome remains to be seen.

## Supporting information

**S1 Checklist.**
(DOCX)

**S1 Annex. Statistical discussion.**
(DOCX)

## Acknowledgments

We are extremely grateful to all the staff at St Mary's Hospital, past and present, who continue to submit data to the NELA database. We would also like to thank the NELA project team for their ongoing support in making national audit data available locally.

## Author Contributions

**Conceptualization:** Michael Berry, Jennifer Louise Gosling.

**Data curation:** Michael Berry, Jennifer Louise Gosling, Rachel Elizabeth Bartlett.

**Formal analysis:** Michael Berry.

**Methodology:** Michael Berry, Stephen James Brett.

**Project administration:** Michael Berry, Rachel Elizabeth Bartlett.

**Supervision:** Stephen James Brett.

**Writing – original draft:** Michael Berry, Stephen James Brett.

**Writing – review & editing:** Michael Berry, Stephen James Brett.

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
