## [Decision Letter · Decision Letter 0]

14 Feb 2022

PONE-D-21-35723Exploring red cell distribution width as a potential risk factor in emergency bowel surgery – a retrospective cohort studyPLOS ONE

Dear Dr. Berry,

Thank you for submitting your manuscript to PLOS ONE. After careful consideration, we feel that it has merit but does not fully meet PLOS ONE’s publication criteria as it currently stands. Therefore, we invite you to submit a revised version of the manuscript that addresses the points raised during the review process.

We look forward to receiving your revised manuscript.

Kind regards,

Itamar Ashkenazi

Academic Editor

PLOS ONE

Journal Requirements:

2. In ethics statement in the manuscript and in the online submission form, please provide additional information about the patient records/samples used in your retrospective study. Specifically, please ensure that you have discussed whether all data/samples were fully anonymized before you accessed them and/or whether the IRB or ethics committee waived the requirement for informed consent. If patients provided informed written consent to have data/samples from their medical records used in research, please include this information.

MB was a Health Service Research Centre fellow with the NELA project form August 2018 – August 2019. All other authors (JLG, REB, SJB) declare that they have no conflict of interest. 

Reviewers' comments:

Reviewer's Responses to Questions

**Comments to the Author**

1. Is the manuscript technically sound, and do the data support the conclusions?

Reviewer #1: Yes

Reviewer #2: Yes

2. Has the statistical analysis been performed appropriately and rigorously? 

Reviewer #1: I Don't Know

Reviewer #2: Yes

3. Have the authors made all data underlying the findings in their manuscript fully available?

Reviewer #1: Yes

Reviewer #2: Yes

4. Is the manuscript presented in an intelligible fashion and written in standard English?

Reviewer #1: Yes

Reviewer #2: Yes

5. Review Comments to the Author

Reviewer #1: Thank you very much for the opportunity to review this interesting research article. The introduction gives a strong basis for the study and its relevance. My few comments ask for some clarification with regards to the methods.

Page 4, line 97 - There should be some justification for the 3 month follow-up period as opposed to 6 months or one year.

Given the NELA risk prediction score is included as part of the model, the components of this should be more explicit for those who are not familiar with this score.

There needs to be some justification for making the ASA score binary and not categorical.

Please give a justification for differentiating major and major + in operative severity and how different surgeries were categorized into each of these areas.

Why not analyze RDW as a continuous variable instead of in quartiles?

Table 1 – it is unclear what the p value of the surgical indication is describing. Please clarify.

Figure 3 – I am unsure what this figure is aiming to show. Maybe there is a better way to compare these variables that is more easy to read? For example, the RDW vs HR is explained, but it is not clear to me how to interpret the surgical indication variables. Is NELA score used as a reference point in this case? Could you also comment on why Cr was used as a variable in the right panel nomogram and not the other variables such as NELA score or hemoglobin?

The discussion is also very focused and well written.

Reviewer #2: This study demonstrates clearly that prognosis of patients undergoing laparascopy is more adverse in the case of a larger red cell distribution width. The authors investigated a large cohort of patients and performed diligent statistical analyses. The study deserves interest and contributes to insights in the clinical impact of red cell distribution.

The study would benefit from an outlook for the future in which way the results of this study could become beneficial for patients in the surgical setting: For instance, identifiying patients at increased risk of peri- and post-surgery complications who may benefit from a closer monitoring in the in- and outpatient setting.

6. PLOS authors have the option to publish the peer review history of their article (what does this mean?). If published, this will include your full peer review and any attached files.

Reviewer #1: No

Reviewer #2: No

---

## [Author Response · Author response to Decision Letter 0]

7 Mar 2022

PLOS ONE style requirement:

Please ensure that your manuscript meets PLOS ONE’s style requirements, including those for file naming. 

The style and file naming have been amended as required. 

2. Ethics statement: 

In the ethics statement in the manuscript and in the online submission form, please provide additional information about the patient records/ samples used in your retrospective study. Specifically, please ensure that you have discussed whether all data/samples were fully anonymised before you accessed them and/or whether the IRB or ethics committee waived the requirement for informed consent. If patients provided informed written consent to have data/samples from their medical records used in research, please include this information. 

Many thanks for this. We have amended the ethics statement as suggested. The added sentence reads: “Under prevailing UK regulations, individual patient consent was not required given the retrospective nature of the. Data were pseudo anonymised using the unique NELA identifier. Handling of online NELA data entered by individual NHS trusts adheres to strict information governance standards, which are laid out on the NELA website [8]. All additional administrative or clinical data required were collected as part of routine clinical care and analysed in accordance with the General Data Protection Regulation.”

3. Competing interests:

Thank you for stating the following in the Competing Interests section: “MB was a Health Service Research Centre fellow with the NELA project form August 2018 – August 2019. All other authors (JLG, REB, SJB) declare that they have no conflict of interest”. Please confirm that this does not alter your adherence to all PLOS ONE policies on sharing data and materials, by including the following statement: "This does not alter our adherence to PLOS ONE policies on sharing data and materials.” (as detailed online in our guide for authors http://journals.plos.org/plosone/s/competing-interests). If there are restrictions on sharing of data and/or materials, please state these. Please note that we cannot proceed with consideration of your article until this information has been declared. Please include your updated Competing Interests statement in your cover letter; we will change the online submission form on your behalf.

Many thanks for highlighting the above. Our revised statement now includes the sentence as suggested. 

MB was a Health Service Research Centre fellow with the National Emergency Laparotomy from August 2018 to August 2019. All other authors (JLG, REB, SJB) declare that they have no conflict of interest. We received no financial support for the research, authorship or publication of this article other than Imperial Biomedical Research Centre infrastructure support for Professor Brett. This does not alter our adherence to PLOS ONE policies on sharing data and materials. 

4. Direct Billing: 

Please note that in order to use the direct billing option the corresponding author must be affiliated with the chosen institute. Please either amend your manuscript to change the affiliation or corresponding author or email us at plosone@plos.org with a request to remove this option.

Thank you for pointing this out, we have amended the corresponding author as requested. The corresponding author will be Professor S. Brett (stephen.brett@imperial.ac.uk).

5. Reference list review: 

Please review your reference list to ensure that it is complete and correct. If you have cited papers that have been retracted, please include the rationale for doing so in the manuscript text or remove these references and replace them with relevant current references. Any changes to the reference list should be mentioned in the rebuttal letter that accompanies your revised manuscript. If you need to cite a retracted article, indicate the article’s retracted status in the References list and also include a citation and full reference for the retraction notice.

Thank you very much for giving us the opportunity to double check our references. We can confirm that none of the articles have been retracted. Furthermore, we have included one additional reference to clarify handling of confidential data in the ethics section (reference number 8) and is marked in red in the reference list.

Reviewer #1: 

1. Thank you very much for the opportunity to review this interesting research article. The introduction gives a strong basis for the study and its relevance. My few comments ask for some clarification with regards to the methods.

Thank you very much for your kind words. 

2. Page 4, line 97 - There should be some justification for the 3 month follow-up period as opposed to 6 months or one year.

The follow-up period was chosen a priori. We looked at 30-day and overall survival after laparotomy. We reviewed data from 1st January 2014 to 31st January 2018 and included all patients during this period. Patients recruited at the beginning of the study naturally have a longer follow-up period compared to patients at the end of the study. Thus, patients with an emergency laparotomy in early 2014 potentially were followed-up for 4 years. Conversely a patient undergoing emergency bowel surgery in late January would have had a minimum follow-up of three months. 

We chose 30-day and overall mortality (with a minimum 3-month follow-up) endpoints for the following reasons. Mortality at 30 days is conventionally accepted to reflect early events. Beyond 30 days death is less likely to reflect periprocedural related mortality. Overall mortality allows the rate of death in this cohort to be examined generally and to pinpoint when it occurs. Examining the cumulative mortality rate plot in Annex 1 (Figure 1) the majority of deaths post emergency laparotomy occur in the first three months. 

Although we accept that a minimum 3-month follow-up period may appear short, most deaths tend to happen in this early phase. Moreover, only a small number of patients (n=21 (6%)) did not have a follow-up period of at least six months (patients operated after the 1st November 2017). We hope that this goes some way to explain our choice of follow-up period. 

3. Given the NELA risk prediction score is included as part of the model, the components of this should be more explicit for those who are not familiar with this score.

We are grateful to the reviewer for highlighting this perfectly reasonable point. We have added the following sentence to our manuscript:” The NELA risk model incorporates routinely collected predictors (patient demographics, physiological data, laboratory values, and perioperative details) and has been published elsewhere [2]. A full summary of the included variables can be found in Annex 1.” The description of the risk factors can be found under Data collection in Annex 1.

4. There needs to be some justification for making the ASA score binary and not categorical.

Although convenient for tabulation and data presentation we agree that dichotomising ASA for these reasons is not justified, particularly as it leads to significant loss of information and statistical power. Hence, we have included all ASA grades in Table 1 as suggested. 

5. Please give a justification for differentiating major and major + in operative severity and how different surgeries were categorized into each of these areas.

We agree that dichotomising operative severity is an oversimplification. The UK National Emergency Laparotomy Audit takes the view that all emergency laparotomies are major procedures, defined as surgery carried out within 24 hours following decision to operate. To nuance a colonic resection from adhesiolysis (division of adhesions causing intestinal obstruction, without resection) the differentiation major and major+ was introduced. This simplified binary categorisation is used across annual NELA reports and in the paper describing the NELA risk model (Eugene et al. Development and internal validation of a novel risk adjustment model for adult patients undergoing emergency laparotomy surgery: The National Emergency Laparotomy Audit risk tool. BJA 2018, 121 (4): 739-748). 

In view of the well-made point made by the reviewer we have clarified the binary categorisation by adding the sentence: “Operative severity was classified according to NELA as major or major+, reflecting surgical immediacy, post-operative length of stay and mortality [1,2]” to the data collection, missing values, and predictor selection paragraph. The footnote below table 1 in the main article details how different surgeries were categorised into each group. 

6. Why not analyze RDW as a continuous variable instead of in quartiles?

Many thanks for this crucial comment, indeed we should have made this clearer. Patient characteristics were examined across RDW quartiles and presented in Table 1. However, all modelling avoided categorisation and RDW was used as a continuous variable throughout. Annex 1 elaborates on this in depth, but we agree that in the main manuscript this is not made obvious enough until the discussion. 

We have amended the sentences describing the models using RDW as a continuous predictor. The sentences are as follows:

“Using RDW as a continuous variable, we went on to evaluate the prognostic value of RDW at predicting mortality outcomes.” [page 6, line 152]

“Finally, to investigate the association between RDW as a continuous variable and frailty, we developed a binary logistic regression model.” [page 7, line 172]

7. Table 1 – it is unclear what the p value of the surgical indication is describing. Please clarify.

Thank you for noting this. We agree that the terminology appears confusing. Surgical indication relates to the surgical pathology requiring a laparotomy. The p value relates to the overall difference in surgical disease across RDW quartiles. We have aligned the diagnostic groupings for greater visual clarity (all are now left justified in the respective cell) and changed surgical indication to surgical pathology. We have detailed the breakdown of surgical pathology in the footnote of Table 1. 

8. Figure 3 – I am unsure what this figure is aiming to show. Maybe there is a better way to compare these variables that is more easy to read? For example, the RDW vs HR is explained, but it is not clear to me how to interpret the surgical indication variables. Is NELA score used as a reference point in this case? 

Many thanks for this valuable feedback. We agree that the left-hand hazard ratio plot is complex. Nonetheless we feel it serves two important purposes. The first is to visually represent estimated predictor effects. The second is to present both point estimates (for surgical indication) and interval estimates for continuous variables (avoid categorisation). The later demonstrates how hazard ratios change for predictors in the model over a range of values. 

We attempted to make this clear in the footnote below the graph but accept that this could be made clearer. Hence we have changed the sentence “Simple HR are presented for categorical predictors” to “Standard HRs are presented for surgical indication. Here the hazard ratio is a conventional comparison of the hazard between two groups.”

9. Could you also comment on why Cr was used as a variable in the right panel nomogram and not the other variables such as NELA score or hemoglobin?

We are grateful for this valuable comment regarding Figure 3. The intention of the nomogram was to depict the model while enabling interactions to be demonstrated. The risk of creatinine is influenced by haemoglobin and the NELA risk score. Thus, using different values for haemoglobin (5th and 90th centile) and NELA risk (25th, 50th and 75th centile) we illustrate how creatinine as a risk predictor is modified. To clarify this, we have changed the footnote below the graph to include the following sentence:” To illustrate this the 5th and 90th centile was chosen for haemoglobin and the interquartile range for the NELA risk score”. We have discussed these interactions in greater depth in Annex 1. 

10. The discussion is also very focused and well written.

Thank you for this kind comment. 

Reviewer #2: 

1. This study demonstrates clearly that prognosis of patients undergoing laparoscopy is more adverse in the case of a larger red cell distribution width. The authors investigated a large cohort of patients and performed diligent statistical analyses. The study deserves interest and contributes to insights in the clinical impact of red cell distribution.

Many thanks for this kind comment. 

2. The study would benefit from an outlook for the future in which way the results of this study could become beneficial for patients in the surgical setting: For instance, identifying patients at increased risk of peri- and post-surgery complications who may benefit from a closer monitoring in the in- and outpatient setting.

Thank you for emphasising the clinical aspect and potential benefit to patients in the future. We agree and have added the following paragraph to strengthen this aspect:

“In turn, superior risk prediction tools could allow more meaningful informed consent and shared decision making between patients and healthcare professionals. At present, it remains unknown whether RDW is a modifiable risk factor perioperatively, including the elective setting. It would be interesting to establish if targeting factors reflected in the RDW improves surgical outcomes. Should tailored interventions such as physical rehabilitation, nutritional support or immunomodulation prove beneficial, this would further strengthen the argument to use RDW to identify individual patients at risk [24].”

---

## [Editor Report · Decision Letter 1]

14 Mar 2022

Exploring red cell distribution width as a potential risk factor in emergency bowel surgery – a retrospective cohort study

PONE-D-21-35723R1

Dear Dr. Brett,

We’re pleased to inform you that your manuscript has been judged scientifically suitable for publication and will be formally accepted for publication once it meets all outstanding technical requirements.

Kind regards,

Itamar Ashkenazi

Academic Editor

PLOS ONE

---

## [Editor Report · Acceptance letter]

14 Apr 2022

PONE-D-21-35723R1 

Exploring red cell distribution width as a potential risk factor in emergency bowel surgery – a retrospective cohort study 

Dear Dr. Brett:

I'm pleased to inform you that your manuscript has been deemed suitable for publication in PLOS ONE. Congratulations! Your manuscript is now with our production department. 

Kind regards, 

on behalf of

Dr. Itamar Ashkenazi 

Academic Editor

PLOS ONE